# Medical service quality, patient satisfaction and intent to revisit: Case study of public hub hospitals in the Republic of Korea

**Selin Woo**, **Mankyu Choi** *

Department of Health Policy & Management, BK21 FOUR R&E Center for Learning Health Systems, College of Health Science, Korea University, Anam-ro, Seongbuk-gu, Seoul, South Korea

* mkchoi@korea.ac.kr

## Abstract

This study aimed to construct and test structural equation modeling of the causal relationship between quality of healthcare, patient satisfaction, and intent to revisit perceived by patients using regional hub public hospitals. In this study, data of 2,951 outpatients and 3,135 inpatients were collected using the "2018 Regional Hub Public Hospital Operational Evaluation." A structural equation model was used to understand the relationship between patient satisfaction and intent to revisit, and bootstrap analysis was performed. In the direct effect, outpatients were presented in the order of the physician's practice service, the hospital's environment, and patient satisfaction. Inpatients were in the order of the physician's practice service and, medical staff's kindness and consideration,; patient satisfaction was shown in this order. In the indirect effect, the outpatients were presented in the order of physician's practice service, medical staff's kindness and consideration, and hospital's physical environment. Inpatients were introduced in the order of medical staff's kindness and consideration, nurse's practice service, physician's practice service, and patient satisfaction. Regional hub public hospitals need high-quality medical services and efforts from all departments to treat patients with sincerity to improve patient satisfaction and increase intent to revisit.

## Introduction

With improved living standards, convenience in access to various information, and rapid societal aging, quantitative and qualitative demands and medical services expectations increase. Additionally, the sharp increase in the number of medical professionals and institutions has made competition between medical institutions inevitable. The increasing level of consciousness and expectations of citizens and the number of medical services consumers mean that only medical institutions meeting these expectations can be managed sustainably. In response to such social trends, the healthcare market is shifting from being supplier-oriented to customer-oriented [1]. Customer-oriented marketing is a standard business administration concept aiming to identify and satisfy customers' needs [2]. The medical industry was previously dominated by a supplier-oriented market where patients visited hospitals without receiving

**Data Availability Statement:** Data cannot be shared publicly because data are owned by a third party. According to Korea National Medical Center Institutional data are protected and cannot be shared publicly. We received permission to access

the data by submitting research proposals to the Korea National Medical Center Institutional. Data can be made available upon request. Individual researchers must submit research proposals and obtain approval when using them. For any data request researchers may contact: Korea National medical center Institutional, Center for public healthcare policy, 251, Eulji-ro, Jung-gu, Seoul; sms@nmc.or.kr, +82 02-6362-3724.

**Funding:** The author(s) received no specific funding for this work.

**Competing interests:** The authors have declared that no competing interests exist.

**Abbreviations:** AVE, average variance extraction; CR, construct reliability.

patient-oriented medical services. However, the customer-oriented market is currently wide-spread because hospitals survive when they understand medical consumers' diverse needs and provide high-quality medical services desired by customers [3]. The essence of customer-oriented marketing in healthcare organizations is to provide quality services to medical consumers. Medical institutions in Korea continue to increase in number. Thus, for the maintenance and survival of healthcare institutions by achieving a competitive advantage, a customer-oriented marketing strategy that satisfies customers' needs by providing quality medical services, leading to revisit, is required.

Healthcare institutions are obligated to provide safe and quality medical services to patients before considering their management and operation. Donabedian [4] divided healthcare quality into "professional skills of the healthcare provider" and "the patient's perception of the served medical service." Bopp [5] also categorized healthcare quality into the medical provider's technical quality and the functional quality perceived by the patient, the medical consumer, and argued that the latter is more important than the former in evaluating healthcare quality. The medical provider-centered quality concerns the level of capacity to provide professional medical skills such as proficiency in medical skills and accurate diagnosis. Contrastingly, the quality perceived by the patients means, in addition to medical skills, the functional quality that indicates the extent to which the patient's demands such as those for facilities, equipment, physical environment, and communication are satisfied [1]. Woolley et al. [6] found that patients may report a "perception of satisfaction" despite a poor medical treatment outcome. In another study by Zifko-Baliga and Krampf [7], if the patient did not recognize that the medical provider provided an accurate diagnosis and treatment, healthcare quality is not high. The quality of healthcare services, in particular, significantly depends on the criteria used in the evaluation by the patients themselves. As patients' needs are diverse and advanced, subjective evaluation based on the patient's position is considered more important in evaluating healthcare quality [8]. Hwang and Shim [9] showed that the quality of healthcare perceived by patients affects their satisfaction and medical services in the future. This means that patient satisfaction, in addition to the quality of healthcare, is significant in terms of hospital profitability. Patient satisfaction with a hospital leads to an intent to revisit the hospitalpublic and private hospitals.

The dramatic increase and development of private hospitals have led to a rapid decrease public hospitals' proportion and role. Recently, the functions and roles of public hospitals have decreased as a result of the avoidance phenomenon due to several reasons: low quality of healthcare failing to meet the expectations of medical consumers, the image that they are old-fashioned, difficulty in obtaining capable medical staff, and the perception that they treat only the vulnerable. They also experience problems in management due to deteriorating financial conditions [10]. Public hospitals are expected to perform functions such as maintaining the medical and social safety net, implementing central and local government policies, and conducting activities for the local community's public interest. Specifically, they provide healthcare services for vulnerable and disabled people, services in maternal-child health (which is in short supply due to low profitability), mental illness, infectious diseases, and emergency care. Public hospitals' maintenance is essential because they are responsible for areas considered challenging to service by private hospitals. However, the de-publicization occurring in the healthcare field, like in other fields, and the decrease in the government's investment in public health led to public hospitals' financial deterioration [11]. Jinju Medical Center is an excellent example of this outcome. The center was established in 1910 and played the role of a public hospital caring for residents' health. However, it closed due to the accumulation of deficits resulting from severe financial deterioration. The decline in the local populations, a phenomenon in which patients are concentrated in large hospitals due to the expansion of medical

insurance coverage in South Korea, and intensification of competition among hospitals are additional factors complicating local medical centers' management [12]. Therefore, the profitability should not be overlooked if the regional hub public hospitals aim to improve residents' health, provide quality medical treatment, and perform the function of a medical safety net that is difficult for private hospitals. Accordingly, the Korean government attempted to establish a system to develop regional hub public hospitals while promoting the public healthcare system's reform. The government's policies to build regional hub public hospitals include the "Comprehensive Plan for the Expansion of Public Health and Medical Care" in 2005, "Plan for Development of Regional Hub Public Hospital" in 2010, "Plan for Improvement of Public Health Care by Development of Regional Medical Center" in 2013, "Master Plan for Public Health Care" in 2016, and "Comprehensive Plan for Development of Public Health Care" in 2018. These policies aimed to re-establish the functions and roles of regional hub public hospitals in the provision of high-quality medical care, medical safety net, and medical care for the vulnerable. If regional hub public hospitals pursue the medical safety net function, which private hospitals do not provid, and profitability, they need to develop measures to increase the revisit rate by improving the quality of medical care and patient satisfaction by reflecting social trends.

Korea's public medical institutions include public health centers, medical institutions for special needs, national university hospitals, and public hub hospitals. Among them, the most crucial role is played by public hub hospitals [13]. Many studies have investigated the effect of quality of medical care service on patient satisfaction and intent to revisit. However, the studies on the use of regional hub public hospitals were scant due to limited available data. There have been no studies, in particular, examining all the regional hub public hospitals across the country as well as both outpatients and inpatients.

## Purpose

This study aimed to investigate the influence of the quality of healthcare perceived by patients who use regional hub public hospitals on their satisfaction and intent to revisit. Based on the results, this study intends to help generate appropriate profits, and perform a medical safety net's function and provide treatment to the vulnerable, which are regional hub public hospitals' goals, by increasing patient satisfaction and intent to revisit. The specific research hypotheses are as follows:

1. In the outpatient case, the physician's practice service, medical staff's kindness and consideration, and the hospital's physical environment will directly or indirectly affect the intent to revisit.

2. In the inpatient case, the physician's practice service, nurse's practice service, the medical staff's kindness and consideration, and the hospital's physical environment will directly or indirectly affect the intent to revisit.

## Materials and method

### Research model

This study measured the direct effect of the quality of healthcare perceived by patients on patient satisfaction and intent to revisit in inpatients and outpatients who used regional hub public hospitals and the indirect impact of healthcare quality on the intent to revisit through the parameter of patient satisfaction. The research model was developed under the assumption established from the literature review that the patient's healthcare quality has a structural causal relationship with patient satisfaction and intent to revisit. In this model, patient satisfaction was a mediating variable reinforcing the relationship between the quality of medical

service as an independent variable and the intent to revisit as a dependent variable. The quality of healthcare was composed of physician's practice service, nurse's practice service, medical staff's kindness and consideration, and the hospital's physical environment. The gender, age, education, and health status of patients were controlled in the analysis.

## Data and participants

This study utilized the patient satisfaction survey results from data collected through the "2018 Evaluation for Operation of Regional Hub Public Hospital" conducted by the Ministry of Health and Welfare and the National Medical Center. Patient satisfaction was divided into outpatient and inpatient satisfaction, and the survey was commissioned to a specialized survey institution. The survey participants were patients over the age of 18 years who used regional hub public hospitals from May 2017 to April 2018. After excluding 1,152 patients who provided no response or incomplete responses 6,086 patients were included in the analysis (response rate: 84.0%), with 2,951 outpatients and 3,135 inpatients. The survey was conducted by investigators of a specialized research survey institution through telephone using a structured self-reported questionnaire.

## Instruments

In this study, healthcare quality, patient satisfaction, and intent to revisit perceived by patients using regional hub public hospitals were measured using a structured questionnaire developed by academic experts, public officials, and public hospital practitioners. The "survey of patient satisfaction" was adapted from the "2018 Evaluation for Operation of Regional Hub Public Hospital" hosted by the Ministry of Health and Welfare and the National Medical Center. The measurement method for each variable is as follows.

**Healthcare service quality.**　The measurement instrument was developed through review and opinion collection by a team of related academics, researchers, and practitioners. The "quality of healthcare service" for the outpatient group was composed of three items: physician's practice service, medical staff's kindness and consideration, and the hospital's physical environment. The "physician's practice service" included questions on appropriate hours of care, medical history check, intelligible explanation, doctor's listening, polite attitude, and professional health care delivery. Further, "medical staff's kindness and consideration" included questions on the attitude of the examination and administration staff, consideration of the medical staff, and description of the reminder. The "hospital's physical environment" included questions on bathroom and, hospital cleanliness and convenience of waiting and auxiliary facilities. For inpaients, "Nurse's practice service" was included, and "quality of medical service" was composed of four items. The nurse's practice service" included questions on polite attitude, nurse's listening, intelligible explanation, quick response to inconveniences, periodic patient check, professional health care delivery, and nurse's satisfaction. The items were answered using a four-point Likert scale (1 = not at all, 2 = sometimes, 3 = mostly, 4 = always) and an 11-point Likert scale. The degree of consistency among items was measured using Cronbach's α [14], which were .901 for "physician's practice service," .821 for "medical staff's kindness and consideration," and .796 for "hospital's physical environment" in the outpatient group. Further, in the inpatient group, the values were .886 for "physician's practice service," .887 for "nurse's practice service," .842 for "medical staff's kindness and consideration," and .824 for "hospital's physical environment."

**Patient satisfaction and intent to revisit.**　The data for patient satisfaction and intent to revisit were extracted from the "Survey of Patient Satisfaction" from the "2018 Evaluation for Operation of Regional Hub Public Hospital." These variables were measured using one item,

which was rated on a scale from 0 to 10. Higher scores indicated higher patient satisfaction and intent to revisit.

## Analysis

First, to verify data normality and identify the participants' demographic characteristics, descriptive statistics and frequency analysis were performed. Second, reliability, correlation, and multicollinearity analyses of the variables were conducted. Third, confirmatory factor analysis was conducted to confirm the model's goodness-of-fit and validity for the structural equation model analysis. Fourth, the structural equation model was used to identify the goodness-of-fit of the final model and the effect of patients' healthcare quality on the relationship between patient satisfaction and intent to revisit. The bootstrap analysis was conducted to test the mediating effect of patient satisfaction. Statistical analyses were performed using SPSS 23.0 and AMOS 24.0 programs.

## Ethical consideration

As this study only uses anonymized secondary data, according to national guidelines, receiving the exemption from Institutional Review Board. This study was conducted after receiving the KUIRB-2020-0124-01 exemption from the Korea University Institutional Review Board.

## Results

### Demographic characteristics of participants

The participants were 6,086 patients including 2,951 outpatients and 3,135 inpatients over the age of 18 who used regional hub public hospitals (regional medical center: n = 34; Red Cross Hospital: n = 5) across the country. Table 1 shows the participants' demographic characteristics. For the outpatients, there were more male patients (n = 1,705; 57.8%) than female patients (n = 1,246; 42.2%). The most common age group was those in the 60s (n = 812; 27.5%),

**Table 1. General characteristics of participants.**

| | | Outpatient | | Inpatient | |
|---|---|---|---|---|---|
| | | N | % | N | % |
| Gender | Male | 1,705 | 57.8 | 1,686 | 53.8 |
| | Female | 1,246 | 42.2 | 1,449 | 46.2 |
| Age(yr) | 18~29 | 204 | 6.9 | 144 | 4.6 |
| | 30~39 | 236 | 8.0 | 153 | 4.9 |
| | 40~49 | 316 | 10.7 | 271 | 8.6 |
| | 50~59 | 622 | 21.1 | 615 | 19.6 |
| | 60~69 | 812 | 27.5 | 822 | 26.2 |
| | ≥70 | 761 | 25.8 | 1,130 | 36.1 |
| Education | ≤Middle school | 968 | 32.8 | 1,469 | 46.9 |
| | High school | 1,094 | 37.1 | 1,005 | 32.1 |
| | ≥College | 889 | 30.1 | 661 | 21.0 |
| Health State | Very good | 284 | 9.6 | 317 | 10.1 |
| | Good | 1,024 | 34.7 | 895 | 28.5 |
| | Moderate | 1,091 | 37.0 | 950 | 30.3 |
| | Poor | 436 | 14.8 | 708 | 22.6 |
| | Very poor | 116 | 3.9 | 265 | 8.5 |
| Total | | 2,951 | 100 | 3,135 | 100 |

followed by those aged over 70 years (n = 761; 25.8%). The most common education level was high school graduation (n = 1,094; 37.1%), followed by under middle school graduation (n = 968; 32.8%) and over college graduation (n = 889; 30.1%). The most common health status perceived by patients was "moderate" (n = 1,091; 37%), followed by "relatively good" (n = 1,024; 34.7%).

For the inpatients, there were more male patients (n = 1,686; 53.8%) than female patients (n = 1,449; 46.2%). The most common age group was those aged over 70 years (n = 1,130; 36.1%), followed by those in their 60s (n = 822; 26.2%). The most common education level was under middle school graduation (n = 1,469; 46.9%), followed by high school graduation (n = 1,005; 32.1%) and over college graduation (n = 661; 21.0%). The most common health status perceived by patients was "moderate" (n = 950; 30.3%), followed by "very bad" (n = 265; 8.5%).

## Goodness-of-fit test of research model

The association analysis among significant variables showed that the association among all latent variables was significant with P<0.01. The multicollinearity problem was not found because the Variance Inflation Factor was under 10. It was under four for all the outpatient group's measured variables and under six for all the inpatient group's measured variables.

The convergence and discriminant validities were tested to determine whether the target concept or attribute was measured. The average variance extraction (AVE) and the concept of construct reliability (CR) were used to test validities [15]. The validities were supported because AVE values were 0.5 or higher and CR values were 0.7 or higher in both the outpatient and inpatient groups [16].

The results of testing the overall structural model used in this study were as follows. For the outpatient group, $\chi2$ = 1303.176, TLI = 0.948, CFI = 0.961, and RMSEA = 0.053 (Table 2). For the inpatient group, $\chi2$ = 3581.292, TLI = 0.939, CFI = 0.949, and RMSEA = 0.055 (Table 3). These results indicate that the indices were within the recommended level, suggesting that the model is suitable.

## Structural model analysis

Fig 1 illustrates the structural model analysis results for the outpatient group. It was found that physician's practice service had a positive (+) significant effect on patient satisfaction ($\beta$ = 0.377, p < .001) and intent to revisit ($\beta$ = 0.243, p < .001). Medical staff's kindness and consideration had a positive (+) significant effect on patient satisfaction ($\beta$ = 0.303, p < .001) but not on intent to revisit ($\beta$ = 0.036, p = .286). The hospitals' physical environment had a

**Table 2. Correlation matrix (outpatient).**

| | Physician's practice service | Medical staff's kindness and consideration | Hospital physical environment | Patient satisfaction | Intent to revisit | AVE | CR |
|---|---|---|---|---|---|---|---|
| Physician's practice service | 1 | | | | | 0.723 | 0.940 |
| Medical staff's kindness and consideration | .687 | 1 | | | | 0.698 | 0.902 |
| Hospital physical environment | .557 | .661 | 1 | | | 0.679 | 0.893 |
| Patient satisfaction | .724 | .615 | .550 | 1 | | | |
| Intent to revisit | .771 | .684 | .613 | .778 | 1 | | |
| $\chi^2$ = 1303.176, TLI = 0.948, CFI = 0.961, RMSEA = 0.053 | | | | | | | |

All coefficient of correlations are significant at the 0.001 level.

**Table 3. Correlation matrix (inpatient).**

|  | Physician's practice service | Nurse's practice service | Medical staff's kindness and consideration | Hospital physical environment | Patient satisfaction | Intent to revisit | AVE | CR |
|---|---|---|---|---|---|---|---|---|
| Physician's practice service | 1 |  |  |  |  |  | 0.799 | 0.960 |
| Nurse's practice service | .615 | 1 |  |  |  |  | 0.736 | 0.951 |
| Medical staff's kindness and consideration | .713 | .703 | 1 |  |  |  | 0.747 | 0.936 |
| Hospital physical environment | .468 | .561 | .619 | 1 |  |  | 0.677 | 0.910 |
| Patient satisfaction | .708 | .720 | .732 | .622 | 1 |  |  |  |
| Intent to revisit | .644 | .585 | .620 | .488 | .737 | 1 |  |  |
| $\chi^2$ = 3581.292, TLI = 0.939, CFI = 0.949, RMSEA = 0.055 | | | | | | | | |

All coefficient of correlations are significant at the 0.001 level.

positive (+) significant effect on patient satisfaction (β = 0.186, p < .001) and intent to revisit (β = 0.049, p = .046). Patient satisfaction had a positive (+) significant effect on the intent to revisit (β = 0.652, p < .001).

The better the physician's and nurse's practice service, medical staff's kindness and consideration, and the hospital's physical environment, the more likely they are to increase patient satisfaction. It was also found that the better the physician's practice service and hospitals' physical environment, the more these were likely to increase patient satisfaction and intent to revisit.

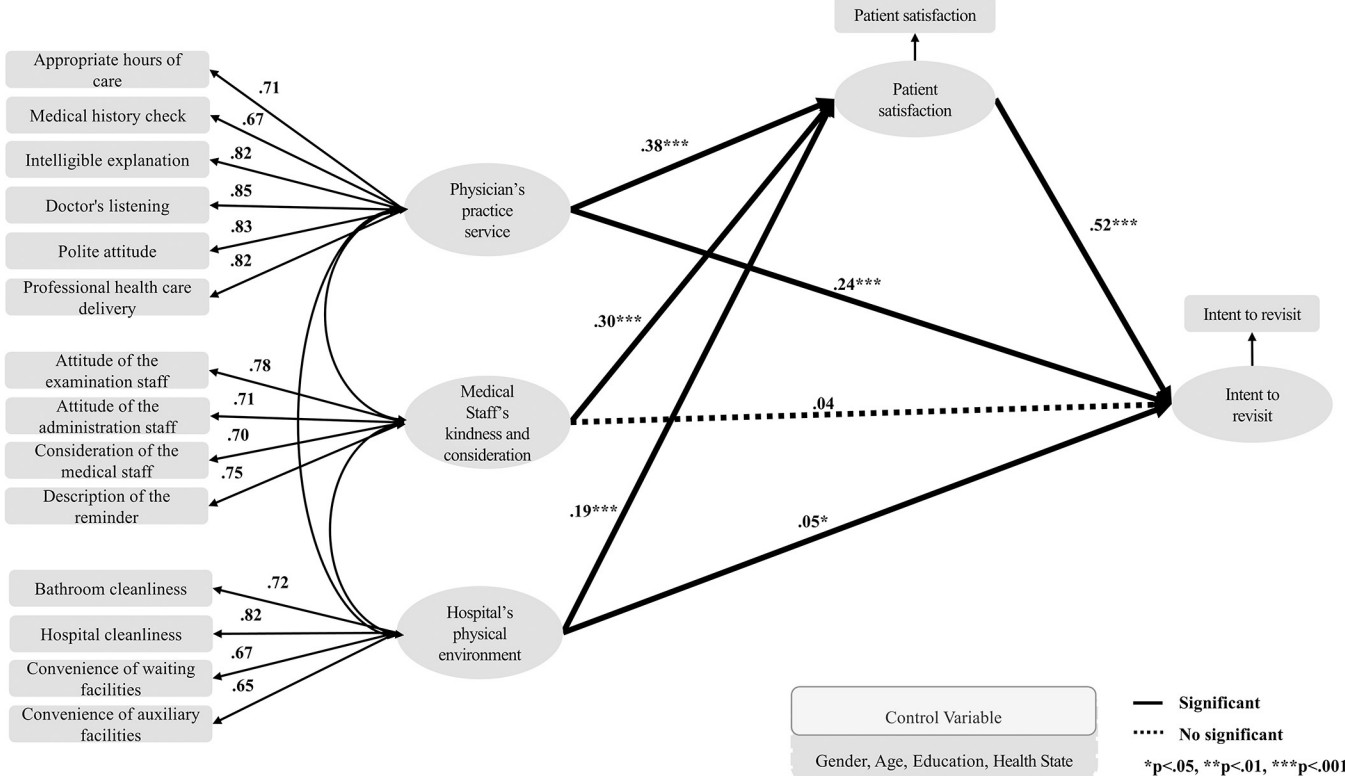

**Fig 1. Path diagram for the structural equation model (outpatient).**

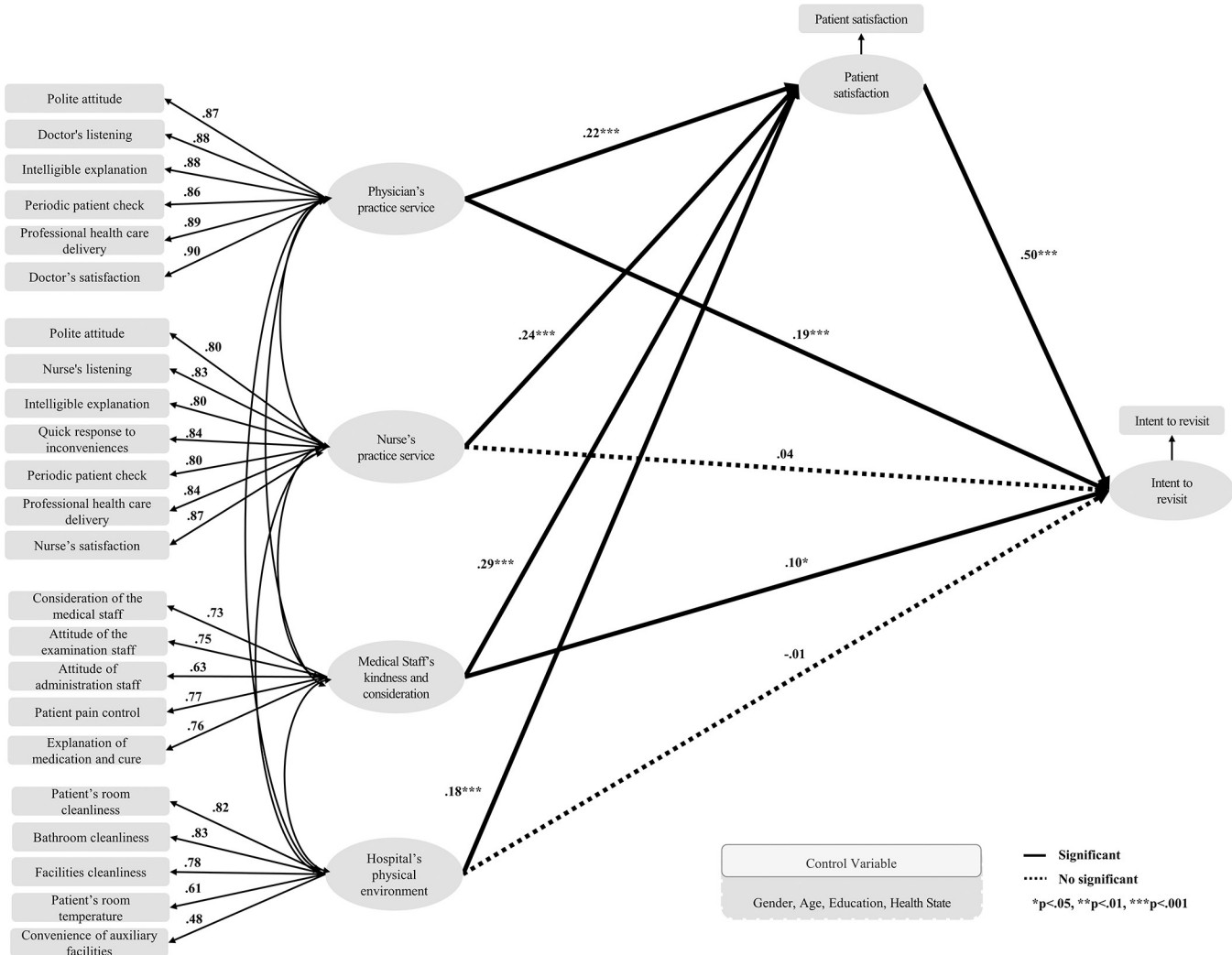

**Fig 2. Path diagram for the structural equation model (inpatient).**

Fig 2 displays the structural model analysis results for the inpatient group. It was found that physician's practice service had a positive (+) significant effect on patient satisfaction ($\beta = 0.223$, $p < .001$) and intent to revisit ($\beta = 0.191$, $p < .001$). Nurses' practice service had a positive (+) significant effect on patient satisfaction ($\beta = 0.236$, $p < .001$) but not on intent to revisit ($\beta = 0.04$, $p = .089$). Medical staff's kindness and consideration had a positive (+) significant effect on patient satisfaction ($\beta = 0.29$, $p < .001$) and intent to revisit ($\beta = 0.096$, $p = .014$). The hospital' physical environment had a positive (+) significant effect on patient satisfaction ($\beta = 0.182$, $p < .001$) but not on intent to revisit ($\beta = -0.013$, $p = .51$). Patient satisfaction had a positive (+) significant effect on the intent to revisit ($\beta = 0.498$, $p < .001$).

Thus, the better the physician's and nurse's practice service, medical staff's kindness and consideration, and the hospital's physical environment, the more likely these are to increase patient satisfaction. It was also found that the better the physician's practice service and the medical staff's kindness and consideration, the more these were likely to increase patient satisfaction and intent to revisit.

## Analysis of the mediating effect

This study used covariance structural analysis to confirm the overall influence among factors as a causal effect. The total, direct, and indirect effects were measured to determine the effect of the independent variable "quality of healthcare" on the dependent variable "intent to revisit," which revealed the importance of the mediating variable "patient satisfaction." The total effect was expressed as the sum of all direct and indirect effects of the independent variable on the dependent variable. The direct effect reflected the direct relationship between the independent and dependent variables; the indirect effect represented the independent variable's effect on the dependent variable through a mediating variable by regression analysis.

Table 4 shows the total, direct, and indirect effects in the outpatient group. The values of the standardized direct effect on "patient satisfaction" were 0.377 for "physician's practice service," 0.303 for "medical staff's kindness and consideration," and 0.186 for "hospital's physical environment," which were statistically significant. The values of the standardized direct effect on "intent to revisit" were 0.243 for "physician's practice service," 0.036 for "medical staff's kindness and consideration," 0.049 for "hospital's physical environment," and 0.522 for "patient satisfaction." Here, the values for "physician's practice service," "hospital's physical environment," and "patient satisfaction" were statistically significant. The values of the standardized indirect effect of the independent variable on the dependent variable through the mediating variable "patient satisfaction" were 0.197 for "physician's practice service," 0.158 for "medical staff's kindness and consideration," and 0.097 for "hospital's physical environment," which were statistically significant. "Patient satisfaction" was found to mediate the relationship between the quality of healthcare perceived by the patient ("physician's practice service," "medical staff's kindness and consideration," and "hospital's physical environment") and intent to revisit.

The direct effect on "intent to revisit" was highest for "patient satisfaction," followed by "physician's practice service," "hospital's physical environment," and "medical staff's kindness and consideration." The indirect effect on "patient satisfaction" through "intent to revisit" was highest for "physician's practice service," followed by "medical staff's kindness and consideration" and "hospital's physical environment." The total effect was highest for "physician's practice service," followed by "medical staff's kindness and consideration," and "hospital's physical environment." These results indicated that "physician's practice service" is the most significant factor influencing "patient satisfaction" and "intent to revisit."

**Table 4. Effect analysis (outpatient).**

| Path | | Direct effect | Indirect effect | Total effect |
|---|---|---|---|---|
| Physician's practice service | Patient satisfaction | 0.377** | - | 0.377** |
| | Intent to revisit | 0.243** | 0.197** | 0.440** |
| Medical staff's kindness and consideration | Patient satisfaction | 0.303** | - | 0.303** |
| | Intent to revisit | 0.036 | 0.158** | 0.195** |
| Hospital's physical environment | Patient satisfaction | 0.186** | - | 0.186** |
| | Intent to revisit | 0.049* | 0.097** | 0.146** |
| Patient satisfaction | Intent to revisit | 0.522** | - | 0.522** |

Bootstrap standardized direct, indirect, total effect.

*p < .05

**p < .01

***p < .001.

**Table 5. Effect analysis (inpatient).**

| Path | | Direct effect | Indirect effect | Total effect |
|---|---|---|---|---|
| Physician's practice service | Patient satisfaction | 0.223** | - | 0.223** |
| | Intent to revisit | 0.191** | 0.111** | 0.302** |
| Nurse's practice services | Patient satisfaction | 0.236** | - | 0.236** |
| | Intent to revisit | 0.040 | 0.117** | 0.158** |
| Medical staff's kindness and consideration | Patient satisfaction | 0.290** | - | 0.290** |
| | Intent to revisit | 0.096* | 0.145** | 0.240** |
| Hospital's physical environment | Patient satisfaction | 0.182** | - | 0.182** |
| | Intent to revisit | -0.013 | 0.090** | 0.077* |
| Patient satisfaction | Intent to revisit | 0.498** | - | 0.498** |

Bootstrap standardized direct, indirect, total effect.

*p < .05

**p < .01

***p < .001.

Table 5 shows the analysis results of total, direct, and indirect effects in the inpatient group. The values of the standardized direct effect on "patient satisfaction" were 0.223 for "physician's practice service," 0.236 for "nurse's practice service," 0.290 for "medical staff's kindness and consideration," and 0.182 for "hospital's physical environment," which were statistically significant. The values of the standardized direct effect on "intent to revisit" were 0.191 for "physician's practice service," 0.040 for "nurse's practice service," 0.096 for "medical staff's kindness and consideration," -0.013 for "hospital's physical environment," and 0.498 for "patient satisfaction" where the values for "physician's practice service," "medical staff's kindness and consideration," and "patient satisfaction" were statistically significant. The values of the standardized indirect effect of the independent variable on the dependent variable through the mediating variable "patient satisfaction" were 0.111 for "physician's practice service," 0.117 for "nurse's practice service," 0.145 for "medical staff's kindness and consideration," and 0.090 for "hospital's physical environment," which were statistically significant. "Patient satisfaction" was found to mediate the relationship between the quality of healthcare perceived by the patient ("physician's practice service," "medical staff's kindness and consideration," and "hospital physical environment") and intent to revisit.

The direct effect on "intent to revisit" was highest for "patient satisfaction," followed by "physician's practice service," "medical staff's kindness and consideration," "nurse's practice service," and "hospital's physical environment." The indirect effect on "patient satisfaction" through "intent to revisit" was highest for "medical staff's kindness and consideration," followed by "nurse's practice service," "physician's practice service," and "hospital's physical environment." The total effect was highest for "physician's practice service," followed by "medical staff's kindness and consideration," "nurse's practice service," and "hospital's physical environment."

## Discussion

This study examined the effect of healthcare quality perceived by patients using regional hub public hospitals on their patient satisfaction and intent to revisit the hospital. A research model was established through a theoretical literature review. To test the model, 6,086 patients, including 2,951 outpatients and 3,135 inpatients over 18 years of age who had experience using regional hub public hospitals were analyzed using the results of the "2018 Evaluation for

Operation of Regional Hub Public Hospital." The analysis used in this study is structural equation modeling. There is also a similar path analysis. Path analysis is a concept first used by Professor Sewall Green Wright [17]. It has the advantage of being able to verify the causal relationship between multiple independent and dependent variables. However, path analysis makes a statistical assumption that there is no measurement error [18]. Such an assumption does not matter if it is measured as a single item. Nevertheless, a problem arises when it is converted into a single item form, such as the average or total score measured by several observation variables [18]. What can solve this problem is structural equation modeling [19]. Structural equation modeling originates from the "JKW model," in which path analysis and confirmatory factor analysis are integrated [20]. Structural equation modeling differs from path analysis and uses the concept of latent variables. Latent variables are not directly observed or measured, but indirectly measured by the observed variable [21]. In other words, this study used structural equation modeling that can use latent variables.

The summarized results are as follows.

First, the variables that directly affected intent to revisit were, for the outpatient group, physician's practice service, hospital's physical environment, and patient satisfaction, and for the inpatient group, physician's practice service, medical staff's kindness and consideration, and patient satisfaction. This is consistent with the results of Kang [22] regarding inpatients in public medical institutions. Kang [22] reported that medical professionalism, the staff's kindness, interest in and service for patients, the convenience of the process of being served in the hospital, and hospital facilities and environment had a positive (+) correlation with intent to revisit. These are consistent with the claims by Yoon [23] and Han [24], respectively, that a higher satisfaction with physician's practice service and hospital environment are likely to increase intent to revisit and that the medical staff's medical service and hospital facility influenced intent to revisit. These results indicate the significance of the physician's treatment skills, which are the essence of medical institutions. Cooperation with private medical institutions is also an excellent way to develop the knowledge and skills of medical personnel working for regional hub public hospitals.

Second, all the variables for the inpatient and outpatient groups had a significant effect on patient satisfaction. The mediating effect on intent to revisit through patient satisfaction was also significant in all variables. This is consistent with the results of Park [25], who investigated the effect of healthcare service quality on intent to revisit and the mediating effect of patient satisfaction in patients who use small and medium hospitals. Park [25] reported that patient satisfaction mediated the relationship between healthcare service quality and intent to revisit. Amarantou et al. [26] investigated the effect of quality of healthcare service on the intent to revisit through the mediating effect of patient satisfaction in emergency room patients in Greece. They reported that patient satisfaction had a positive (+) significant mediating effect. These results suggest that it is necessary to develop various strategies to enhance the quality of medical care services and continuously increase patient satisfaction to improve patients' intent to revisit.

The effect analysis results indicated that the indirect effect is highest, in the outpatient group, for physician's practice service, followed by medical staff's kindness and consideration, and the hospital's physical environment. Further, in the inpatient group, the indirect effect is highest for medical staff's kindness and consideration, followed by nurse's practice service, physician's practice service, and the hospital's physical environment. The results showed that, among healthcare quality factors, medical staff's kindness and consideration had the most substantial effect in outpatient and inpatient groups. Park [25] had investigated the effect of quality of healthcare on intent to revisit through patient satisfaction and had results consistent with those of this study, reporting that the indirect effect was highest in the empathy factor,

which indicates a consideration for customers, among the quality of medical services factors. Aliman and Mohamad [27] showed that, among the quality of medical service factors, the staff's ability and the confidence representing goodwill, respectful attitude, and sincere interest in patients had the most potent effect on patient satisfaction and intent to revisit. The difference in the most critical factor between outpatient and inpatient groups of this study is due to the practice process. Outpatients are likely to leave the hospital just after seeing the physician; therefore, the physician's practice is most important. However, unlike outpatients, inpatients are present in the hospital for medical services. This allows them many opportunities to communicate with hospital employees. For the customer's maintenance, the medical service provider should ensure that quality healthcare service, which is the essence of healthcare, is provided. The medical staff show kindness, sincerity, care, and respect for patients to be encouraged to revisit. For example, Hospital A ranked 1st in the patient experience evaluation conducted by the Health Insurance Review and Assessment Service. The most frequently mentioned factor was the staff's kindness toward patients. Hospital B, which ranked 2nd in the evaluation, implemented the "Activity as Patient for Experience," in which employees experienced the patient's treatment process to find a space for improvement to create a patient-oriented medical culture in their hospital. These examples demonstrate that the provision of a service that considers the patient's position influences patient satisfaction. If regional hub public hospitals seek to provide quality healthcare to the public while maintaining and surviving in the current strong competition, developing programs to educate medical staff and all staff dealing with patients to treat patients sincerely is needed. Regular management of patients through active training and monitoring by the quality assurance team is also necessary.

This study has several limitations. First, there is an insufficient understanding of the causal relationship between variables. This is a cross-sectional study using data from the "2018 Evaluation for Operation of Regional Hub Public Hospital." Thus, future studies using longitudinal data are recommended to clarify how healthcare quality affects patient satisfaction and intent to revisit. Second, this study did not include factors that may affect patient satisfaction and intent to revisit due to data restrictions on personal information protection and confidentiality, consultation of doctors, nurses, or designees, especially in inpatients with major diseases. In the future, research will need to proceed with the study, including other factors that may affect patient satisfaction and intent to revisit.

Nevertheless, this study contributed to this field in that it investigated the management strategy to generate appropriate profits for regional hub public hospitals by analyzing the relationship among quality of healthcare, patient satisfaction, and intent to revisit. Previous studies have been conducted mainly targeting private hospitals, in outpatients and inpatients using regional hub public hospitals after controlling for gender, age, health status, and education level.

## Conclusion

This study used a structural equation model to examine whether there is a relationship among patient's healthcare service quality, patient satisfaction, and intent to revisit in outpatients and inpatients visiting regional hub public hospitals. The study aimed to provide fundamental data to improve patient satisfaction and service in regional medical centers and Red Cross hospitals.

The results indicated that, in the outpatient group, the better the physician's practice service, hospital's physical environment, and patient satisfaction, the more likely these were to increase the patients' intent to revisit. The indirect effect on intent to revisit through patient satisfaction was highest for physician's practice service, followed by medical staff's kindness

and consideration, and the hospital's physical environment. Moreover, in the inpatient group, the better the physician's practice service, medical staff's kindness and consideration, and patient satisfaction, the more these were likely to increase the patients' intent to revisit. The indirect effect on intent to revisit through patient satisfaction was highest for medical staff's kindness and consideration, followed by nurse's practice service, physician's practice service, and patient satisfaction.

Based on the results, we present the following suggestions. First, public hospitals' environment should be improved to satisfy medical consumers to strengthen publicity through fiscal consolidation. Second, providing active healthcare service through cultivating professional knowledge and technical expertise in healthcare professionals should be attempted. Finally, the medical service provider should ensure the provision of quality healthcare service and that medical staff show kindness, sincerity, care, and respect for patients so that they are encouraged to revisit.

## Author Contributions

**Conceptualization:** Selin Woo.

**Data curation:** Selin Woo.

**Formal analysis:** Selin Woo.

**Methodology:** Selin Woo.

**Resources:** Selin Woo.

**Supervision:** Mankyu Choi.

**Validation:** Mankyu Choi.

**Writing – original draft:** Selin Woo.

**Writing – review & editing:** Mankyu Choi.

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
