## [Decision Letter · Decision Letter 0]

18 Mar 2021

PONE-D-21-00964

The effect of quality of medical service on patient satisfaction and intent to revisit: for public hub hospitals in Republic of Korea

PLOS ONE

Dear Dr. Choi,

Thank you for submitting your manuscript to PLOS ONE. After careful consideration, we feel that it has merit but does not fully meet PLOS ONE’s publication criteria as it currently stands. Therefore, we invite you to submit a revised version of the manuscript that addresses the points raised during the review process.

Reviewers felt that the paper needs minor editing for clarity. You may use the services of a professional editor to improve the readability of the paper. You should also address few additional substantive issues raised by the reviewers.

A rebuttal letter that responds to the important points raised by the reviewer(s). You should upload this letter as a separate file labeled 'Response to Reviewers'.A marked-up copy of your manuscript that highlights changes made to the original version. You should upload this as a separate file labeled 'Revised Manuscript with Track Changes'.An unmarked version of your revised paper without tracked changes. You should upload this as a separate file labeled 'Manuscript'.

We look forward to receiving your revised manuscript.

Kind regards,

M. Mahmud Khan

Academic Editor

PLOS ONE

Journal Requirements:

3) Please include additional information regarding the survey or questionnaire used in the study and ensure that you have provided sufficient details that others could replicate the analyses. For instance, if you developed a questionnaire as part of this study and it is not under a copyright more restrictive than CC-BY, please include a copy, in both the original language and English, as Supporting Information. Moreover, please include more details on how the questionnaire was pre-tested, and whether it was validated.

4) We note that you have indicated that data from this study are available upon request. PLOS only allows data to be available upon request if there are legal or ethical restrictions on sharing data publicly. For information on unacceptable data access restrictions, please see http://journals.plos.org/plosone/s/data-availability#loc-unacceptable-data-access-restrictions.

5) Your ethics statement should only appear in the Methods section of your manuscript. If your ethics statement is written in any section besides the Methods, please move it to the Methods section and delete it from any other section. Please ensure that your ethics statement is included in your manuscript, as the ethics statement entered into the online submission form will not be published alongside your manuscript.

6) Please upload a copy of Supporting Information Figures S1 & S2 and Tables S1, S2 and S3 which you refer to in your text on page 19.

Reviewers' comments:

Reviewer's Responses to Questions

**Comments to the Author**

1. Is the manuscript technically sound, and do the data support the conclusions?

Reviewer #1: Partly

Reviewer #2: Yes

2. Has the statistical analysis been performed appropriately and rigorously? 

Reviewer #1: Yes

Reviewer #2: Yes

3. Have the authors made all data underlying the findings in their manuscript fully available?

Reviewer #1: Yes

Reviewer #2: Yes

4. Is the manuscript presented in an intelligible fashion and written in standard English?

Reviewer #1: No

Reviewer #2: Yes

5. Review Comments to the Author

Reviewer #1: 1. Research purpose should be listed as hypotheses, such as hypothesis 1, 2, etc.

2. Professional translation is recommended. Many sentences and paragraphs have grammatical errors and are not clear.

3. Clear definition on variables are suggested. What is 'practice service'? What are the difference between 'physician's practice service' and 'nurce's practice service'? Staff's consideration?

4. Also define 'regional public hub hospital in Korea'. Based on function, work scope, and size of the hospitals, user group could be vary. Could you generalize the results of this study to all public hospitals in Korea?

5. Recommend to use a table to describe results of the Goodness-of-fit test (page 9).

6. (Page 10) This is 'Path analysis', not 'Structural Equation Modeling'.

7. (Page 16) Authors described, 'This study has several limitations', but list one limitation. Is there any other limitation in this study?

Reviewer #2: The paper is successful in its attempt and addressed its objectives well. However, it may also shed lights on some other issues related to patient’s satisfaction. This may not be necessarily related to the quality of healthcare as perceived by the patients, such as some local issues related to the local culture and tradition, practices, etc. Issues of privacy and confidentiality, counselling by doctors, nurses or designated, especially for the inpatients suffering from some major ailments, etc. are also major factors which affect the satisfaction of the patients and their intension to revisit the public health facilities. The problem of overcrowding and shortfalls of medical and paramedical staff especially in the developing country is also a major challenge faced by the public health facilities.

6. PLOS authors have the option to publish the peer review history of their article (what does this mean?). If published, this will include your full peer review and any attached files.

Reviewer #1: No

Reviewer #2: **Yes: **Rajeev K Kumar

---

## [Author Response · Author response to Decision Letter 0]

13 Apr 2021

Reviewer 1:

Comment 1. Research purpose should be listed as hypotheses, such as hypothesis 1, 2, etc

Response 1: Thank you for your comment. In reflection of the reviewer’s opinion we added the following to the to the research purpose (page 5).

“The specific research hypotheses are as follows: 

1. In the outpatient case, the physician’s practice service, medical staff’s kindness and consideration, and the hospital’s physical environment will directly or indirectly affect the intent to revisit. 

2. In the inpatient case, the physician’s practice service, nurse’s practice service, medical staff’s kindness and consideration, and the hospital’s physical environment will have a direct or indirect effect on the intent to revisit.”

Comment 2. Professional translation is recommended. Many sentences and paragraphs have grammatical errors and are not clear.

Response 2: Thank you for your comment. In reflection of the reviewer’s opinion, we revised the manuscript for the English language.

Comment 3. Clear definition on variables are suggested. What is 'practice service'? What are the difference between 'physician's practice service' and nurse's practice service'? Staff's consideration?

Response 3: Thank you for your comment. Based on the reviewer’s opinion, the following sentence has been added to the Healthcare Service Quality section in Materials and Methods (page 7):

The “physician’s practice service” included questions on appropriate hours of care, medical history check, intelligible explanation, doctor’s listening, polite attitude, and professional health care delivery. Further, “medical staff’s kindness and consideration” included questions on the attitude of the examination and administration staff, consideration of the medical staff, and description of the reminder. The “hospital’s physical environment” included questions on bathroom and hospital cleanliness and convenience of waiting and auxiliary facilities. For inpatients, “nurse’s practice service” was included, and “quality of medical service” was composed of four items. The “nurse’s practice service” included questions on polite attitude, nurse’s listening, intelligible explanation, quick response to inconveniences, periodic patient check, professional health care delivery, and nurse’s satisfaction. 

Figures 1 and 2 refer to the following questions about the physician’s practice service, nurse’s practice service, medical staff’s kindness and consideration, and the hospital’s physical environment. You can see that the physician’s practice service is focused on clinical medical services, and the nurse’s practice service includes social services. Medical staff’s kindness and consideration include a question about whether the medical staff had sufficient consideration so that the patient’s body was not exposed during the treatment or examination, such as exposure to the patient’s body; therefore, the variable name was set as “medical staff’s kindness and consideration.”

The questions for each item were regarding the following:

- physician’s practice service: appropriate hours of care, medical history check, intelligible explanation, doctor’s listening, polite attitude, professional health care delivery

- nurse’s practice service: polite attitude, nurse’s listening, intelligible explanation, quick response to inconveniences, periodic patient check, professional health care delivery, nurse’s satisfaction

- medical staff’s kindness and consideration: attitude of the examination and administration staff, consideration of the medical staff, description of the reminder

- hospital’s physical environment: bathroom and hospital cleanliness, convenience of waiting and auxiliary facilities

Comment 4. Also define 'regional public hub hospital in Korea'. Based on function, work scope, and size of the hospitals, user group could be vary. Could you generalize the results of this study to all public hospitals in Korea?

Response 4: Thank you for the comments. Based on the reviewer’s opinion, the following sentence has been added to the Introduction section (page. 4):

Korea’s public medical institutions include public health centers, medical institutions for special needs, national university hospitals, and public hub hospitals. Among them, the most crucial role is played by public hub hospitals [13]. 

13. PUBLIC HEALTH AND MEDICAL SERVICES ACT https://law.go.kr/engLsSc.do?menuId=1&subMenuId=21&tabMenuId=117&query=%EA%B3%B5%EA%B3%B5%EB%B3%B4%EA%B1%B4#

Public health institutions in Korea include public health centers, public hub hospitals (public hub hospitals, Red Cross hospitals), national university hospitals, and medical institutions for special needs (police, industrial accidents, veterans, military hospitals, etc.). Among them, public hospitals with regional bases are the core of Korean public health. They are established to provide public health care to residents, for whom private health care cannot be performed, and private health care cannot be performed in vulnerable areas. The data used in this study were targeted at all 39 public hub hospitals (34 local medical centers, 5 Red Cross hospitals). The results of this study cannot be generalized to all public hospitals. However, as samples were extracted from all 39 public hub hospitals, which are hospitals in regional units distributed across the country, generalization is possible for public hub hospitals.

Comment 5. Recommend to use a table to describe results of the Goodness-of-fit test (page 9).

Response 5: Thank you for your comment. In response to the reviewer's opinion, the following table has been added to the fitness test on pages 10-11.

Comment 6. (Page 10) This is 'Path analysis', not 'Structural Equation Modeling'.

Response 6: Thank you for your comment. Based on the reviewer’s comment, Figures 1 and 2 are modified and presented in the text (page 12). Further, the following sentence has been added to the discussion section (page 16).

“The analysis used in this study is structural equation modeling. There is also a similar path analysis. Path analysis is a concept first used by Professor Sewall Green Wright [17]. It has the advantage of being able to verify the causal relationship between multiple independent and dependent variables. However, path analysis makes a statistical assumption that there is no measurement error [18]. Such an assumption does not matter if it is measured as a single item. Nevertheless, a problem arises when it is converted into a single item form, such as the average or total score measured by several observation variables [18]. What can solve this problem is structural equation modeling [19]. Structural equation modeling originates from the “JKW model,” in which path analysis and confirmatory factor analysis are integrated [20]. Structural equation modeling differs from path analysis and uses the concept of latent variables. Latent variables are not directly observed or measured but indirectly measured by the observed variable [21]. In other words, this study used structural equation modeling that can use latent variables.”

17. WRIGHT, Sewall. The method of path coefficients. The annals of mathematical statistics, 1934, 5.3: 161-215.

18. James B. Schreiber, Amaury Nora, Frances K. Stage, Elizabeth A. Barlow & Jamie King (2006) Reporting Structural Equation Modeling and Confirmatory Factor Analysis Results: A Review, The Journal of Educational Research, 99:6, 323-338, DOI: 10.3200/JOER.99.6.323-338

19. CHEUNG, Gordon W.; LAU, Rebecca S. Testing mediation and suppression effects of latent variables: Bootstrapping with structural equation models. Organizational research methods, 2008, 11.2: 296-325.

20. Bentler, P. M. Multivariate analysis with latent variables: Causal modeling. Annual review of psychology. 1980;31(1), 419-456.

21. BOLLEN, Kenneth A. Latent variables in psychology and the social sciences. Annual review of psychology, 2002, 53.1: 605-634.

Comment 7. (Page 16) Authors described, 'This study has several limitations', but list one limitation. Is there any other limitation in this study?

Response 7: Thank you for your comment. One more limit has been added to reflect reviewer 2’s opinion in the Discussion section (page 18).

“Second, this study did not include factors that may affect patient satisfaction and intent to revisit due to data restrictions on personal information protection and confidentiality, consultation of doctors, nurses, or designees, especially in inpatients with major diseases. In the future, research will need to proceed with the study, including other factors that may affect patient satisfaction and intent to revisit.”

Reviewer 2:

Comment 1. The paper is successful in its attempt and addressed its objectives well. However, it may also shed lights on some other issues related to patient’s satisfaction. This may not be necessarily related to the quality of healthcare as perceived by the patients, such as some local issues related to the local culture and tradition, practices, etc. Issues of privacy and confidentiality, counselling by doctors, nurses or designated, especially for the inpatients suffering from some major ailments, etc. are also major factors which affect the satisfaction of the patients and their intension to revisit the public health facilities. The problem of overcrowding and shortfalls of medical and paramedical staff especially in the developing country is also a major challenge faced by the public health facilities.

Response 1: Thank you for your comment. Based on the reviewer's opinion, the following sentence has been added to the Discussion section (page 18):

“Second, this study did not include factors that may affect patient satisfaction and intent to revisit due to data restrictions on personal information protection and confidentiality, consultation of doctors, nurses, or designees, especially in inpatients with major diseases. In the future, research will need to proceed with the study, including other factors that may affect patient satisfaction and intent to revisit.”

---

## [Editor Report · Decision Letter 1]

12 May 2021

Medical service quality effect on patient satisfaction and intent to revisit: For public hub hospitals in the Republic of Korea

PONE-D-21-00964R1

Dear Dr. Choi,

We’re pleased to inform you that your manuscript has been judged scientifically suitable for publication and will be formally accepted for publication once it meets all outstanding technical requirements.

Kind regards,

M. Mahmud Khan

Academic Editor

PLOS ONE

Additional Editor Comments (optional):

There are still some awkward sentences in the manuscript and needs editing. In my opinion, the title needs changes as well. I suggest that the title of the paper be changed to reflect the content of the paper. A possible title could be, "Medical service quality, patient satisfaction and intent to revisit: Case study of public hub hospitals in the Republic of Korea". 
---

## [Editor Report · Acceptance letter]

17 May 2021

PONE-D-21-00964R1 

Medical service quality, patient satisfaction and intent to revisit: Case study of public hub hospitals in the Republic of Korea 

Dear Dr. Choi:

I'm pleased to inform you that your manuscript has been deemed suitable for publication in PLOS ONE. Congratulations! Your manuscript is now with our production department. 

Kind regards, 

on behalf of

Dr. M. Mahmud Khan 

Academic Editor

PLOS ONE